# EEG Frequency Correlates with α_2_-Receptor Density in Parkinson’s Disease

**DOI:** 10.3390/biom14020209

**Published:** 2024-02-10

**Authors:** Adam F. Kemp, Martin Kinnerup, Birger Johnsen, Steen Jakobsen, Adjmal Nahimi, Albert Gjedde

**Affiliations:** 1Department of Nuclear Medicine, Odense University Hospital, 5000 Odense, Denmark; adam_felix_kemp@hotmail.com; 2Department of Clinical Medicine, Aarhus University, 8000 Aarhus, Denmark; m3kinnerup@gmail.com (M.K.); birgjohn@rm.dk (B.J.); steejako@rm.dk (S.J.); 3Department of Clinical Neurophysiology, Aarhus University Hospital, 8200 Aarhus, Denmark; 4Clinical Memory Research Unit, Department of Clinical Sciences, 211 46 Malmö, Sweden; adjmalnahimi@gmail.com; 5Department of Neurology, Skåne University Hospital, 221 85 Lund, Sweden; 6Department of Neuroscience, University of Copenhagen, 1172 Copenhagen, Denmark; 7Department of Neurology and Neurosurgery, McGill University, Montreal, QC H3A 0G4, Canada; 8Translational Neuropsychiatry Unit, Department of Clinical Medicine, Aarhus University, 8000 Aarhus, Denmark

**Keywords:** Parkinson’s, dementia, EEG, α_2_ adrenoceptor, noradrenaline, locus coeruleus

## Abstract

Introduction: Increased theta and delta power and decreased alpha and beta power, measured with quantitative electroencephalography (EEG), have been demonstrated to have utility for predicting the development of dementia in patients with Parkinson’s disease (PD). Noradrenaline modulates cortical activity and optimizes cognitive processes. We claim that the loss of noradrenaline may explain cognitive impairment and the pathological slowing of EEG waves. Here, we test the relationship between the number of noradrenergic α_2_ adrenoceptors and changes in the spectral EEG ratio in patients with PD. Methods: We included nineteen patients with PD and thirteen healthy control (HC) subjects in the study. We used positron emission tomography (PET) with [^11^C]yohimbine to quantify α_2_ adrenoceptor density. We used EEG power in the delta (δ, 1.5–3.9 Hz), theta (θ, 4–7.9 Hz), alpha (α, 8–12.9 Hz) and beta (β, 13–30 Hz) bands in regression analyses to test the relationships between α_2_ adrenoceptor density and EEG band power. Results: PD patients had higher power in the theta and delta bands compared to the HC volunteers. Patients’ theta band power was inversely correlated with α_2_ adrenoceptor density in the frontal cortex. In the HC subjects, age was correlated with, and occipital background rhythm frequency (BRF) was inversely correlated with, α_2_ adrenoceptor density in the frontal cortex, while occipital BRF was inversely correlated with α_2_ adrenoceptor density in the thalamus. Conclusions: The findings support the claim that the loss or dysfunction of noradrenergic neurotransmission may relate to the parallel processes of cognitive decline and EEG slowing.

## 1. Introduction

Non-motor symptoms are very frequent in patients with Parkinson’s disease (PD) and may present prior to motor symptoms. One of the most debilitating and prevalent non-motor symptoms in PD is cognitive impairment. More than 30% of patients with PD may have mild cognitive impairment early in the disease and these patients have a significantly increased risk of developing PD dementia [1].

The underlying pathophysiology of cognitive impairment in PD patients is complex and not completely understood. Neuropathological studies suggest that the widespread accumulations of misfolded alpha-synuclein in cortical, limbic and brainstem structures play a significant role in the cognitive impairment of PD patients. In addition, amyloid and tau co-pathologies become more significant as PD patients progress to express dementia. The pathological accumulation of alpha-synuclein in the brain is associated with neuronal deterioration and the loss of key neurotransmitters, including acetylcholine and noradrenaline, that lead to cognitive deterioration [2,3].

Noradrenergic terminals of fibers that arise from the locus coeruleus and reach most areas of the brain, except the striatum, modulate alertness, attention, executive functions, working memory and the sleep–wake cycle [4]. The loss of noradrenaline occurs early in PD patients and may play a significant role in cognitive impairment, a hypothesis that arose from previous studies of PD patients with rapid eye movement (REM) sleep behavior disorder. REM sleep behavior disorder (RBD) among PD patients has been linked to the more pronounced loss of NA terminals and a higher number of non-motor symptoms of PD, including a higher rate of cognitive impairment in parallel with slowed resting EEG activity [5]. The EEG markers that correlated with dementia in PD patients are the slowing of the background rhythm frequency (BRF) and the increased power in the theta and delta bands [6,7].

Noradrenaline exerts its effects through α_1_, α_2_ and β adrenoceptors. In animal studies, α_2_ adrenoceptor antagonism raises the EEG spectral ratio (SR) value, defined as the sum of the power in the alpha and beta bands, divided by the sum of the power in the theta and delta bands. This effect of α_2_ adrenoceptor antagonism may be due to an increase in noradrenaline levels, to which α_2_ adrenoceptors act as auto-receptors on presynaptic noradrenaline terminals, in support of which claim α_2_ adrenoceptor agonists have the opposite effect [8]. In a recent study, Sommerauer and co-workers showed that PD patients with RBD have a more extensive noradrenergic deficiency as well as more a pronounced slowing of EEG, i.e., lower alpha/theta ratios [5]. The studies show that deficits of noradrenergic neurotransmission may slow EEG in parallel with the cognitive impairment. However, previous studies did not test the correlation between α_2_ adrenoceptor density and changes in EEG and in the cognitive impairment of PD patients. To the best of our knowledge, no previous studies have measured changes in EEG power during or after challenge with α_2_ adrenoceptor compounds in PD patients. However, one study of patients with attention deficit hyperactivity disorder showed that the administration of guanfacine, an α_2a_ adrenoceptor agonist, reduced the power in the alpha band, as expected from the animal studies mentioned above [9]. The effects of α_2_ adrenoceptors on cognitive function may be dose dependent. Smaller doses of α_2_ adrenoceptor agonists may slow reaction time and reduce vigilance and alertness because of the activation of presynaptic α_2_ adrenoceptors that lower synaptic noradrenaline levels. However, at higher doses, α_2_ adrenoceptor agonists that are sufficient to activate postsynaptic α_2_ adrenoceptors may improve indices of reactivity, vigilance and attention [10]. Thus, an intact noradrenaline system is vital for cognitive function, alertness and vigilance.

We showed previously that the density of α_2_ adrenoceptors can be quantified by imaging with [^11^C]yohimbine positron emission tomography (PET) [11]. Interestingly, a recently published study showed α_2_ adrenoceptors, as measured with [^11^C]yohimbine, are diminished in PD patients, in correlation with both motor and non-motor indices [12]. In the present study, we hypothesize that the degeneration of the LC by the loss of noradrenergic terminals is associated with the pathological slowing of EEG patterns in PD patients. We tested the hypothesis by using [^11^C]yohimbine PET to reveal the relationship between α_2_ adrenoceptor densities and the pathological slowing of EEG frequencies in PD patients.

## 2. Materials and Methods

### 2.1. Participants

The Ethics Committee of Central Denmark Region approved the study. All subjects provided written informed consent prior to participation. We recruited patients with PD from the Department of Neurology at Aarhus University Hospital, Denmark, and healthy control (HC) volunteers by newspaper advertisements. We examined all participants for major comorbidity, and we screened for dementia. Patients fulfilled the UK Parkinson’s Disease Society Brain Bank diagnostic criteria for Parkinson’s disease [13], and we screened them with the Mini-Mental State Examination (MMSE) [14] to exclude dementia by excluding patients with a score below 24. Major psychiatric comorbidity, diagnosed according to the Diagnostic and Statistical Manual of Mental Disorders IV-TR (DSM IV-TR), served as an exclusion criterion. We screened the HC volunteers for neurological disorders, which served as exclusion criteria. With these procedures and criteria, we recruited 19 patients with PD (13 males) aged 53–80 years, with a mean age of 65 years, and 13 HC volunteers aged 52–76 years (6 males), with a mean age of 66 years. Of the participants, 17 of the 19 with PD and all 11 HCs underwent [^11^C]yohimbine PET.

### 2.2. Radiochemistry

We implemented the synthesis of [^11^C]yohimbine as described elsewhere [15], with only minor modifications in this study. Briefly, we converted cyclotron-produced [^11^C]dioxide to [^11^C]methyliodide that we trapped in DMSO (300 μL) containing NaOH (1 μL, 3 mol/L) and yohimbinic acid (1 mg) in a 1-mL vial. We heated the mixture at 80 °C for 3 min. We purified [^11^C]yohimbine by semipreparative high-performance liquid chromatography (HPLC). We delivered the mobile phase, consisting of 75% aqueous 70 mmol/L Na_2_HPO_4_ and 25% ethanol, at a rate of 5 mL/min to a LUNA C18(2) (Phenomenex, Torrance, CA, USA, 250 × 10 mm^2^) semipreparative column with online radio and visible ultraviolet (UV, 280 nm) detection. We collected the fraction containing [^11^C]yohimbine (retention time, 7–8 min) and diluted it with 5 mL sterile saline, filtered through a sterile 0.22 μm filter to obtain a total of 10 mL product solution. This procedure gave 1–2 GBq [^11^C]yohimbine with radiochemical purity in excess of 99% in a sterile formulation ready for injection. We injected mean radioactivity doses of 412 (±51) MBq and 409 (±42) MBq at baseline in patients with PD and HC volunteers, respectively.

### 2.3. Positron Emission Tomography

Before the PET imaging, subjects paused L-DOPA medication for at least 12 h and paused direct dopamine agonists for more than 24 h. We evaluated the severity of motor symptoms with the Unified Parkinson’s Disease Rating Scale (UPDRS) Part 3 in the “off” state. We used the PET imaging protocol for [^11^C]yohimbine, described and validated in a previous study from this group [11]. All participants reclined in the ECAT High Resolution Research Tomograph (HRRT, CTI/Siemens, Knoxville, TN, USA), with their heads immobilized by a custom-built head-holder. We obtained 6 min transmission scans and 90 min dynamic PET scans, consisting of 28 frames with increasing duration (8 × 15 s, 4 × 30 s, 6 × 60 s, 4 × 300 s, 6 × 600 s), the latter recorded in list mode upon administration of a bolus [^11^C]yohimbine.

### 2.4. Image Processing

We used the Ordered Subset Expectation Maximization 3D Ordinary Poisson (OSEM 3D-OP) algorithm including point spread function modeling, with 10 iterations and 16 subsets, for reconstruction of a 256 × 256 × 207 data volume at a resolution of 1.8 mm at full width at half maximum. We coupled the PET images with T1 weighted magnetic resonance imaging (MRI) by the Siemens MAGNETOM Trio with PMOD v.3.5 software and its module PNEURO (PMOD Technologies Ltd., Zurich, Switzerland) for model-based image co-registration and segmentation of the volumes of interest (VOIs). We averaged frames 5–25 of the dynamic PET recording to provide sufficient anatomical detail, and we registered the individual anatomical PET images to individual anatomical MR images by rigid co-registration with a mutual information algorithm. We automatically outlined VOIs on the MR images according to the Hammers maximum probability atlas, implemented in PMOD [16]. We visually corrected the VOI outlines and placements when necessary. We calculated parametric images of the volumes of distribution (mL/cm^3^) by means of the Logan plot [17] with metabolite-corrected tracer blood concentrations as input function. We created parametric images of the binding potentials relative to the non-displaceable binding (BP_ND_) with the module PXMod in PMOD, according to the Logan reference tissue model with corpus callosum as reference region. We completed the kinetic analysis of the [^11^C]yohimbine exchange between circulation and tissue accumulation with the one-tissue compartment model of the processes of unidirectional clearance (mL/cm^3^/min) and efflux rate (min^−1^) of the tracer. We calculated the binding potentials of [^11^C]yohimbine from the kinetics as previously described [11]. To calculate the value of the binding capacity *B*_max_, we used the values of *K*_D_ reported by Phan et al. [18] for the reshaped Eadie–Hofstee version of the Michaelis–Menten equation, *B*_max_ = *B* + *K*_D_ BP_ND_, assuming the absolute binding (*B*) to be negligible in both the PD patients and HC volunteers, as no unlabeled ligand was added to the injectates. We used the equation to estimate the receptor availability (*B*_max_–*B*) and completed the regression analyses with *B*_max_ as the dependent variable for negligible *B*.

### 2.5. EEG

On a subsequent day, we completed EEG records with 19 leads distributed based on the International 10/20 System, with the right mastoid as the ground for all participants. We used the data from baseline EEG recordings to create epochs of 30 s with the patients awake with closed eyes. We manually inspected the epochs for artifacts and we rejected artifactual epochs. We passed the epochs through a 10% cosine window, processed them with fast Fourier transform (FFT) and averaged them to produce an FFT power spectrum for each electrode. We defined the frequencies of the background rhythm (BRF) as the dominant peaks in the power spectra of the occipital electrodes (O1 and O2) identified by visual inspection. We defined the frequency bands as follows: delta (δ) 1.5–3.9 Hz, theta (θ) 4–7.9 Hz, alpha (α) 8–12.9 Hz and beta (β) 13–30 Hz. We used all the electrodes but Fp1 and Fp2 to calculate the power. We defined the spectral ratios as the power in the alpha + beta bands divided by the power in the delta + theta bands.

### 2.6. Statistics

We used the Statistical Package for Social Science (SPSS) version 24 for all the statistical analyses. For the group comparisons, we used the chi-squared, *t*-test, ANOVA and Mann–Whitney tests. We established the correlation between the $B_{max}$ values and EEG patterns by linear regression with stepwise removal of the least significant explanatory at each step. Explanatories eliminated early were reintroduced to check for confounding. Each analysis began with the inclusion of the variables of age, gender, band power of all the bands, background rhythm frequency (BRF) and spectral ratios as explanatories. We removed variables due to the limited number of participants in our study, but did not remove variables having a significant impact. We removed variables from the analyses in cases of multi-collinearity. We plotted the correlation between significant variables to compare patterns between the patients with PD and the HC volunteers. We tested the normality of the data with the Shapiro–Wilk test and the homoscedasticity by Levene’s test and the manual inspection of the scatter plots of residuals. For the unmet assumptions underlying the tests, we analyzed the data with the Mann–Whitney test. We considered a probability of less than 0.05 to be non-random (significant), with use of Bonferroni correction for multiple analyses.

## 3. Results

Table 1 lists the demographics and comparisons of the patients with PD and the HC volunteers. On the day of the PET, the patients had UPDRS scores of 41 (range 17–55) in the “off” state with overnight medication withdrawn, a mean Hoehn and Yahr stage 2.5 (range 2–3) and mean L-dopa equivalent daily doses (LEDD) averaging 1052 mg (range 360–2057 mg). Patients with Parkinson´s disease had more power in the delta and theta bands but no significant reduction in BRF in the occipital electrodes, and we found no significant differences in *B*_max_ or BP_ND_ between the PD patients and HC subjects (Table 1).

Table 2 and Table 3 show the complete results of the regression analyses of the effects of age, gender, band power, BRF and spectral ratios on [^11^C]yohimbine binding in the frontal cortex and thalamus in the PD patients and HCs. We used linear regression to test the relationship between α_2_ adrenoceptor density and the specific parameters of the EEG. We found that α_2_ adrenoceptor density (*B*_max_) in the frontal cortex had an inverse correlation with theta power in PD patients (Table 2). In the HC volunteers, the α_2_ adrenoceptor *B*_max_ in the frontal cortex was correlated with age and BRF, while the α_2_ adrenoceptor *B*_max_ in the thalamus correlated with BRF (Table 3).

The left panel of Figure 1 shows the comparison between HC volunteers and PD patients in terms of theta power. The right panels of the figure show the correlations between theta power and α_2_ adrenoceptor *B*_max_ estimates in the frontal cortex and thalamus. The *p*-values shown in the right panels apply to the PD patients. We plotted the HC volunteers in the same figures but found no significant correlations between theta power and α_2_ adrenoceptor *B*_max_ values among the HC volunteers.

## 4. Discussion

To the best of our knowledge, this is the first study of the direct correlates of EEG alterations with the binding capacity of α_2_ adrenoceptors in the human brain. We tested the claim that α_2_ adrenoceptor density may correlate with EEG patterns. Interestingly, we found negative correlations between the EEG theta band power and the α_2_ adrenoceptor densities, as measured with [^11^C]yohimbine binding potential (receptor availability) in the frontal cortex in patients, and between the BRF and [^11^C]yohimbine binding potential in the frontal cortex and thalamus in the HC volunteers. The inverse correlations indicate that α_2_ adrenoceptors in part are responsible for the forming of EEG patterns in general, as linked to the slowing of the EEG.

### 4.1. Encephalography

EEG is a measure of the fluctuations of field potentials resulting from ionic currents within the neurons responsible for the neurotransmission underlying the neural networks. The changes in EEG among patients with PD appear as a slowing of EEG frequencies, which becomes more evident with the progression of PD [19], with an increase in the power of low-frequency bands, a decrease in the power of high-frequency bands and a lowering of the spectral ratios (SRs) [20]. The changes are associated with cognitive impairments and with the risk of developing dementia as a complication of PD [21]. We did not find that patients and HC volunteers differ in terms of the power of the high-frequency bands that may relate to the exclusion of patients with MMSE values below 24. If the lower power in high-frequency bands is a sign of a link to cognitive impairment, the limit possibly excluded this group of patients. This possibility is supported by the findings of Caviness et al. [21] of a relative decrease in the power of the high-frequency bands relative to the cognitive decline of patients with PD. It also agrees with Bosman et al. [22], who argued that synchronization and, by extension, the power of the high-frequency gamma band support multiple cognitive processes.

Patients had more power in the low-frequency bands compared to the healthy control volunteers, in agreement with earlier reports. Caviness et al. [23] found that higher EEG delta band power was correlated with more widespread synucleinopathy, indicating more severe disease. Klassen et al. [6] found that higher theta band power predicted dementia in patients with PD, and Caviness et al. [21] found that delta and theta band power increased in parallel with greater cognitive decline. It is possible that the high power of low-frequency bands may be a protective mechanism for cognitive processes when the locus coeruleus (LC) degenerates, which is supported by the evidence that states that low-frequency bands share an association with activation of the LC in rodents [24]. LC firing has also been proposed to synchronize and increase the power of high-frequency bands [25].

### 4.2. Binding of [^11^C]yohimbine

In a recent study of [^11^C]MeNER binding to noradrenaline transporters, patients with PD had reduced [^11^C]MeNER binding in the LC compared to HC volunteers [5]. We found no differences of α_2_ adrenoceptor densities, measured with tracer [^11^C]yohimbine, between the patients with PD and the HC volunteers. The reason may be that α_2_ adrenoceptors reside both pre- and postsynaptically. When α_2_ adrenoceptor density decreases presynaptically due to LC degeneration, postsynaptic α_2_ adrenoceptor density may rise to maximize the effect of the available NA. The compensation may conceal the LC degeneration when total α_2_ adrenoceptor availability is a measured proxy. The mechanism may be similar to that of dopamine D_2_ receptors, which initially show increased density in patients with PD, possibly as compensation for dopaminergic denervation.

The EEG measures correlated with the binding potential of [^11^C]yohimbine, both for HC volunteers and for patients with PD. As α_2_ adrenoceptor expression is a measure of the cellular function of certain brain cells, and EEG is an external measure of brain function, it is of interest that the EEG alterations may be the direct effects of altered α_2_ adrenoceptor expression, as found in animals, since α_2_ adrenoceptor antagonists increase spectral ratios in rodents [8].

The α_2_ adrenoceptors are likely to be implicated in the pathophysiology of neurodegenerative and neuropsychiatric disorders [26,27,28]. NA is essential to the regulation of attention [29], and α_2_ adrenoceptor agonists have a complex effect on attention, with older studies showing that nonspecific α_2_ adrenoceptor agonists may reduce attention [30,31,32,33,34,35], while antagonists increase attention in repetition tasks and reverse the attention-reducing effects of agonists [33,36]. However, more recent studies suggest that compounds that are specific α_2a_ adrenoceptors may improve attention and are used as a treatment of attention deficit hyperactivity disorder (ADHD). Similarly, atomoxetine, which is a noradrenaline reuptake inhibitor, is also an approved treatment for ADHD. In addition, NA released by the activation of the LC contributes to mechanisms that optimize or modulate learning, memory and cognitive processes, the sleep–wake cycle, levels of arousal, olfaction, vigilance, regulation of attention, the processing of sensory information in all modalities, cognitive flexibility and long-term synaptic plasticity [25,29,37,38,39]. The links may in part explain how the NA denervation of the LC may relate to global cognitive performance in patients with PD, as shown by means of [^11^C]MeNER imaging [5].

### 4.3. Dementia

The finding of a correlation between the decline of the binding potential of [^11^C]yohimbine and the increase in theta band power among patients with PD is of interest, as high theta power among patients has been linked to cognitive impairment [7,21], aging [40] and an increased risk of developing dementia [6]. The higher power in low-frequency bands, and the low BRF, may be the results of mechanisms that protect the cognitive functions of the individual when the synchronization of high-frequency bands depends on lower frequency bands [25] and cognitive functions depend on the synchronization of high-frequency bands [22]. These associations may explain how changes in EEG may relate to cognitive dysfunction in patients with PD [41], with more pronounced changes following the progression of dementia in the patients [19]. We did not find spectral ratios to be correlated with the binding potentials of [^11^C]yohimbine; the lack of correlation was possibly caused by a protective mechanism that upregulates the number of postsynaptic α_2_ adrenoceptors.

The current findings point to specific cellular processes that may lead to the cognitive decline in PD patients. The inverse correlation between the power of the EEG theta band and the binding potential of [^11^C]yohimbine in principle relates molecular changes to pathological changes in EEG. A possible correlation between the power in the theta band and the binding potential of [^11^C]yohimbine is consistent with the hypothesis that the degeneration of the LC and deficient NA transmission may contribute to joint alterations in EEG and cognitive impairment in PD patients. The joint alterations are clinically relevant, as the combination may reflect the pathology underlying the subgroup of patients suffering from PD, as well as being a key to alternative therapy options for this subgroup. With respect to the subgroups of patients, the present study did not permit the extended following of patients after imaging after the onset of PD. As subgroups are likely to differ in the course of disease, it is important to know the duration of the disease when paraclinical data are compared.

### 4.4. Limitations

The study is limited by the low number of participants, which may explain the lack of strong correlation between the value of *B*_max_ measured in the thalamus and the power in the theta band registered in patients. As the design of the study allowed only the search for correlation and not for causation, we believe that future studies should be designed to increase the likelihood of identifying causation and maybe also to more precisely identify subgroups of patients with a risk of developing cognitive impairments. The impact of the study is limited also by the complexity of the relationships among neurotransmitters, because the correlation between EEG and α_2_ adrenoceptor expression may not be straightforward and may include multiple mechanisms and neurotransmitters that we cannot in any way rule out in the present study. 

## 5. Conclusions

The present study reveals a negative correlation between EEG theta band power and α_2_ adrenoceptor expression in patients suffering from PD. The correlation showing theta band power to be associated with α_2_ adrenoceptor expression supports the hypothesis of the importance of α_2_ adrenoceptors in the generation of EEG patterns. The lack of difference between the HC volunteers and the patients with PD in terms of [^11^C]yohimbine binding suggests that the correlation between LC degeneration and PD is not straightforward and that LC degeneration may be linked to certain subtypes of PD.

The findings are an important step towards a better understanding of the role that noradrenergic neurotransmission plays in the pathophysiology of PD. By linking EEG measures of global functions of the brain to measures of α_2_ adrenoceptor densities, the study provides new insights into the underlying mechanisms of cognitive impairment in PD patients and lays the foundation for future studies. We propose that future studies should be longidutinal and measure several parts of noradrenaline neurotransmission, including the density of α_2_ adrenoceptors and noradrenaline transporters as an indirect measure of surviving neurons, in parallel to longitudinal measurements of cognitive function. 

## Figures and Tables

**Figure 1 biomolecules-14-00209-f001:**
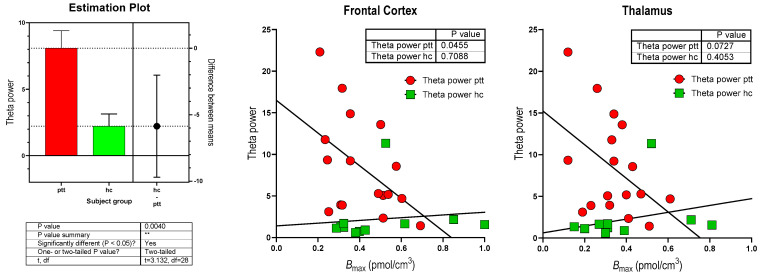
**Comparison of theta power between HC volunteers and patients with PD.** Comparison of theta power of HC volunteers and patients suffering from PD measured in terms of µV^2^. The *p*-values shown in the right panels apply to PD patients, where ptt refers to patients with PD, hc to HC subjects. ** indicates the *p* value summary.

**Table 1 biomolecules-14-00209-t001:** Characteristics of HC subjects and patients with PD.

Variables	PD Mean (Std. Deviation)	HC Mean (Std. Deviation)	*p*-Value
Gender	Male = 12	Male = 6	0.696
	Female = 5	Female = 5	
Age (years)	65.39 (8.70)	66.55 (8.24)	0.726
Delta Power (µV^2^)	3.10	1.23	0.002 *
Theta Power (µV^2^)	8.08	2.21	0.002 *†
Alpha Power (µV^2^)	9.20	8.47	0.450 †
Beta Power (µV^2^)	3.09	3.15	1.000 †
Total Power (µV^2^)	23.47	15.06	0.128 †
BRF (Hz)	7.75 (1.26)	8.28 (1.03)	0.252
Spectral Ratio	1.61	3.47	0.021 *†
Thalamus BP_ND_ (ratio)	0.68 (0.26)	0.78 (0.42)	0.452
Frontal Cortex BP_ND_ (ratio)	0.82	0.99	1.000 †
Thalamus *B*_max_ (pmol/cm^3^)	0.34 (0.13)	0.39 (0.21)	0.452
Frontal Cortex *B*_max_ (pmol/cm^3^)	0.41	0.49	1.000 †

* Indicates *p* ≤ 0.005. † We applied Mann–Whitney U test, as data failed to be normally distributed. BRF refers to background rhythm frequency, V to volt. BP_ND_ refers to binding potential, non-displaceable.

**Table 2 biomolecules-14-00209-t002:** Factors correlated with α_2_ adrenoceptor density in patients with PD.

*B*_max_ Frontal Cortex		*p*-Value	*B*_max_ Thalamus		*p*-Value
Variables	Coefficients		Variables	Coefficients	
Gender	0.232	0.4127	Gender	0.127	0.683
Age	0.255	0.290	Age	0.173	0.509
BRF (Hz)	−0.466	0.145	BRF (Hz)	−0.439	0.208
Theta Power (µV^2^)	−0.620	0.022 *	Theta Power (µV^2^)	−0.550	0.055
			R^2^ = 0.426, adjusted R^2^ = 0.234		

* Indicates *p* ≤ 0.05. BRF refers to background rhythm frequency, V to volt.

**Table 3 biomolecules-14-00209-t003:** Factors correlated with α_2_ adrenoceptor density in HC subjects.

*B*_max_ Frontal Cortex		*p*-Value	*B*_max_ Thalamus		*p*-Value
Variables	Coefficients		Variables	Coefficients	
Gender	0.927	0.080	Gender	0.793	0.120
Age	0.640	0.039 *	Age	0.540	0.119
BRF (Hz)	−1.491	0.021 *	BRF (Hz)	−1.262	0.046 *
Spectral Ratio	0.624	0.228	Spectral Ratio	0.312	0.483
Beta (µV^2^)	0.031	0.902	Delta (µV^2^)	0.257	0.479
			R^2^ = 0.687, adjusted R^2^ square = 0.374		

* Indicates *p* ≤ 0.05. BRF = background rhythm frequency.

## Data Availability

Due to data sharing regulations in Denmark (GDPR) it is not possible to share the data used in this study.

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
