# Peer review of "EEG Frequency Correlates with α2-Receptor Density in Parkinson’s Disease"

_biomolecules, 2024, doi:10.3390/biom14020209_

Round 1
Reviewer 1 Report
Comments and Suggestions for Authors
Major issues:
1. The introduction does not detail the authors hypotheses. If the work was exploratory, this is fine but it should be stated and they should still elaborate their expected observations, in terms of the direction of effect and specific EEG markers of interest alongside the PET quantification measure they intended to use.
2. Background for the EEG measures chosen is lacking in the introduction. The authors mention SR values with reference to noradrenergic activity in animals but do not mention any other measures in animals or humans. The introduction should summarise the involvement (or lack thereof) of noradrenergic signalling for the measures reported in this paper, in particular those that are significantly associated with yohimbine PET signal in their investigation, namely theta power, delta power, SR.
4. A scatter plot (or other appropriate graph) showing the distribution of Bmax values within all participants would be useful given the lack of significant group differences between patients and controls. Is the lack of difference due to variation in the measure across the sampled population or due to limits of sample size and statistical testing? Is there visual evidence of group divergence? Similar graphs could be offered for EEG measures but are less needed.
5. Following on from point 4, the discussion does not seem to engage with the uncertainty generated by the lack of between group differences. Why do the authors associate EEG changes to a2 receptor density if there is also not evidence of group differences in a2 density? This may be reasonable but should be addressed and considered openly and even-handedly.
6. More broadly the limitations of this paper are barely discussed. A paragraph in the discussion entitled strengths and limitations would be a clear way to do this. Please consider these and write accordingly.
7. Some points in the discussion are speculative and without more robust support from the literature, appear as 'just so' explanations for unexplained associations from a regression analysis. Please improve the clarity and focus of the discussion in line with points 5-7.
8. Suggestions for future work should be made clearer and should follow on from the above.
Minor issues:
English language editing needed for minor 'sharpening'
The statement in the abstract "... predict the risk of dementia..." is both grammatically incorrect and overstates the current certainty of evidence regarding the utility of EEG in dementia diagnosis. Better to say something like - " ... has been demonstrated to have utility for predicting the development of/phenoconversion to/onset of dementia OR has been used to identify individuals at risk of developing dementia..
In the methods, the authors state that the PET imaging protocol is used as described elsewhere with minor modifications but do not clearly elaborate on these modifications - this should be done, no need to be coy.
Comments on the Quality of English Language
There are a few typos or minor English language errors. I would suggest review by a native/fluent speaker to sharpen this up.
Examples:
Abstract - "increased theta and delta powers" - should be "power" only
See minor comment above
"We claim that loss of noradrenaline partly may explain the cognitive impairments and pathological slowing of EEG" - sounds clunky, phrasing and style would not be typical of English-speaking scientific writer.
"subjects refrained from intake of medication overnight"
Author Response
Reviewer 1:
Major issues:
- The introduction does not detail the authors hypotheses. If the work was exploratory, this is fine but it should be stated and they should still elaborate their expected observations, in terms of the direction of effect and specific EEG markers of interest alongside the PET quantification measure they intended to use.
Author response:
Thank you for this comment. We agree that the hypothesis of our study should be stated more clearly in the introduction. We have added this and also added a more background information for the hypothesis.
Changes to manuscript: Page 2, third paragraph, line 7: “The EEG markers that correlated with dementia in PD patients are the slowing of the background rhythm frequency (BRF) and the increased powers of the theta and delta bands (6,7).”
And page 3, line 9: “we hypothesize that the degeneration of the LC by loss of noradrenergic terminals is associated with pathologically slowing of EEG patterns in PD patients. We tested the hypothesis by using…”
- Background for the EEG measures chosen is lacking in the introduction. The authors mention SR values with reference to noradrenergic activity in animals but do not mention any other measures in animals or humans. The introduction should summarise the involvement (or lack thereof) of noradrenergic signalling for the measures reported in this paper, in particular those that are significantly associated with yohimbine PET signal in their investigation, namely theta power, delta power, SR.
Author response: We thank the reviewer for this comment and agree that the introduction should be adjusted accordingly, and we added the following:
Changes to manuscript: Page 2 ca line 4 from bottom: “To the best of our knowledge, no previous studies measured changes of EEG power during or after challenge with α2 adrenoceptor compounds in PD patients. However, one study of patients with attention-deficit-hyperactivity disorder showed that administration of guanfacine, an α2a adrenoceptor agonist, reduced the power in the alpha-band as expected from the animal studies mentioned above (9). The effects of α2 adrenoceptors on cognitive function may be dose dependent. Smaller doses of α2 adrenoceptor agonists may slow reaction time and reduce vigilance and alertness because of the activation of presynaptic α2 adrenoceptors that lowers synaptic noradrenaline levels. However, at higher doses, α2 adrenoceptor agonists that are sufficient to activate postsynaptic α2 adrenoceptors may improve indices of reactivity, vigilance and attention (10). Thus, an intact noradrenaline system is vital for cognitive function, alertness, and vigilance.”
- A scatter plot (or other appropriate graph) showing the distribution of Bmax values within all participants would be useful given the lack of significant group differences between patients and controls.Is the lack of difference due to variation in the measure across the sampled population or due to limits of sample size and statistical testing? Is there visual evidence of group divergence? Similar graphs could be offered for EEG measures but are less needed.
Author response: We agree that the Bmax values are of special interest. This is also why we do plot the distribution of Bmax values within all participants in both thalamus and frontal cortex in figure 1 using theta power as the independent value. We believe that the visual distribution provides an insight into a possible difference in patterns between groups that would provide inspiration for future research.
- Following on from point 4, the discussion does not seem to engage with the uncertainty generated by the lack of between group differences. Why do the authors associate EEG changes to a2 receptor density if there is also not evidence of group differences in a2 density? This may be reasonable but should be addressed and considered openly and even-handedly.
Author response: We thank the reviewer for pointing out the observation that average Bmax values of thalamus and frontal cortex do not differ between the two groups as listed in Table 1. We do find differences of theta power as shown in the left panel of Figure 1. However, when we depict theta powers as functions of the estimates of Bmax, as shown in the right panels of the revised Figure 1, we find the evidence of significant or near-significant correlations between Bmax estimates and theta powers in patients, unlike healthy volunteers who display no correlation between theta powers and Bmax estimates. We have revised figure 1 to be clearer on this point. In the revised version we have used Bmax as the independent variable as this is the relation we are trying to elucidate.
Changes to manuscript: Page 11 at figure 1:
“Figure1.
“
- More broadly the limitations of this paper are barely discussed. A paragraph in the discussion entitled strengths and limitations would be a clear way to do this. Please consider these and write accordingly.
Author response: Thank you for this comment, we have added a discussion of limitations in the manuscript
Changes to manuscript: Page 9, line 5: ““With respect to subgroups of patients, the present study did not permit extended following of patients after imaging after onset of PD. As subgroups are likely to differ in the course of disease, it is important to know the duration of disease when paraclinical data are compared”.
Limitations
The study is limited by the low number of participants that may explain the lack of strong correlation between the value of Bmax measured in thalamus and the power of theta registered in patients. As the design of the study only allowed the search for correlation and not for causation, we believe that future studies should be designed to increase the likelihood of causation and maybe also to more precisely identify subgroups of patients with risk of developing cognitive impairments. The impact of the study is limited also by the complexity of the relations among neurotransmitters because the correlation between EEG and α2 adrenoceptor expression may not be straightforward and may include multiple mechanisms and neurotransmitters that we cannot in any way rule out in the present study.”
- Some points in the discussion are speculative and without more robust support from the literature, appear as 'just so' explanations for unexplained associations from a regression analysis. Please improve the clarity and focus of the discussion in line with points 5-7.
Author response: “Thank you for pointing out this weakness. We regard the Discussion as a forum for introduction of hypothetical explanations of novel or not previously discussed findings, including the interpretation of correlations that do not readily suggest specific mechanisms of interaction. We have softened the wording in these cases, as indicated in the resubmitted version“.
- Suggestions for future work should be made clearer and should follow on from the above.
Author response: We have further specified suggestions for future work in the revised version.
Changes to manuscript: Page 9, line 13: “As the design of the study only allowed the search for correlation and not for causation, we believe that future studies should be designed to increase the likelihood of causation and maybe also to more precisely identify subgroups of patients with risk of developing cognitive impairments.”
And on page 10, line 8: “By linking EEG measures of global functions of the brain to measures of α2 adrenoceptor densities, the study provide new insights to the underlying mechanisms of cognitive impairment in PD and lays the foundation for future studies. We propose that future studies should be longidutinal and measure several parts of noradrenaline neurotransmission, including the density of α2 adrenoceptors and noradrenaline transporters as an indirect measure of surviving neurons in parallel to longitudinal measurements of cognitive function.”
Minor issues:
English language editing needed for minor 'sharpening'
The statement in the abstract "... predict the risk of dementia..." is both grammatically incorrect and overstates the current certainty of evidence regarding the utility of EEG in dementia diagnosis. Better to say something like - " ... has been demonstrated to have utility for predicting the development of/phenoconversion to/onset of dementia OR has been used to identify individuals at risk of developing dementia.
Author response: Thank you, we agree that this statement may overstate the certainty of evidence. We have changed the phrasing of the predictive value of EEG.
Changes to manuscript: See page 1, “Abstract”, line 2: “has demonstrated to have utility for predicting the development of dementia in patients with Parkinson’s disease (PD).”
In the methods, the authors state that the PET imaging protocol is used as described elsewhere with minor modifications but do not clearly elaborate on these modifications - this should be done, no need to be coy.
Author response: We agree with the reviewer that the descripition of the PET imaging protocol should be adjusted. In the initial study of [11C]yohimbine PET we validated the method and we employed the method in the current study. The sentence in the methodology is therefore altered to the following:
Changes to manuscript: Page 4, line 5 from bottom: “We used the PET imaging protocol for [11C]yohimbine, described and validated in a previous study from this group.(11)”
Comments on the Quality of English Language
There are a few typos or minor English language errors. I would suggest review by a native/fluent speaker to sharpen this up. Examples:
Abstract - "increased theta and delta powers" - should be "power" only
Author response: Thank you. However, “theta AND delta” are two powers that require plural. Please note that the senior author is an anglophone Canadian citizen.
No changes to manuscript: See page 1, “Abstract”, line 1: “Increased theta and delta powers and decreased alpha and beta powers, measured…”
But we did make the change at line 9: “We used EEG power in the delta (δ) 1.5–3.9”
See minor comment above
"We claim that loss of noradrenaline partly may explain the cognitive impairments and pathological slowing of EEG" - sounds clunky, phrasing and style would not be typical of English-speaking scientific writer.
Author Response: Thank you, although we do not agree entirely, we have in places rephrased some of these texts to follow the advice of the reviewer and to be more clear.
Changes to manuscript: Page 1, “Abstract”, line 4: “We claim that loss of noradrenaline may explain the cognitive impairments and pathological slowing of EEG”
"subjects refrained from intake of medication overnight"
Author Response: Thank you, there was no added information in these words. We have chosen to remove them.
Reviewer 2 Report
Comments and Suggestions for Authors
Cognitive impairment in Parkinson disease (PD) is one of most important non-motor symptom of PD. In one third of PD patients there is a mild cognitive impairment and there is a significant risk of development of PD related dementia.
PD dementia could be caused by accumulation of misfolded alfa-synuclein in cerebral cortex, to amyloid and tau-pathologies and also could be a neurotransmitters involvement (acetylcholine, noradrenaline).
The authors explore the theory that the loss of noradrenaline could be responsible for cognitive impairment and slowing of EEG activity (slowing of EEG frequencies becomes more evident with progression of PD).
They used positron emission tomography (PET) with [11C] yohimbine to quantify α2 adreno-receptors density and quantitative electroencephalography (EEG) to evaluate EEG spectral ratio (SR) value (power of the alpha and beta bands, divided by the power of the theta and delta bands) in 19 PD patients and 13 normal subjects.
They found that PD patients had higher powers of the theta and delta bands compared to helathy volunteers and α2 adrenoceptor density in frontal cortex had an inverse correlation with theta power in PD patients.
Author Response
Thank you very much for reviewing this paper. We will also upload the answers to all comments
Reviewer 2:
Comments and Suggestions for Authors
Cognitive impairment in Parkinson disease (PD) is one of most important non-motor symptom of PD. In one third of PD patients there is a mild cognitive impairment and there is a significant risk of development of PD related dementia.
PD dementia could be caused by accumulation of misfolded alfa-synuclein in cerebral cortex, to amyloid and tau-pathologies and also could be a neurotransmitters involvement (acetylcholine, noradrenaline).
The authors explore the theory that the loss of noradrenaline could be responsible for cognitive impairment and slowing of EEG activity (slowing of EEG frequencies becomes more evident with progression of PD).
They used positron emission tomography (PET) with [11C] yohimbine to quantify α2 adreno-receptors density and quantitative electroencephalography (EEG) to evaluate EEG spectral ratio (SR) value (power of the alpha and beta bands, divided by the power of the theta and delta bands) in 19 PD patients and 13 normal subjects.
They found that PD patients had higher powers of the theta and delta bands compared to helathy volunteers and α2 adrenoceptor density in frontal cortex had an inverse correlation with theta power in PD patients.
Reviewer 3 Report
Comments and Suggestions for Authors
GENERAL POINTS
The correlation of noradrenergic system markers with EEG measures in both PD and healthy subjects is interesting and promising. The correlations currently presented are overall marginal and not adequate to strongly favour the presented hypotheses regarding a significant role of the noradrenergic system on EEG spectral characteristics. Certain modifications of the manuscript may help.
While Table 1 suitably contains all variables of interest, Tables 2 and 3 (Correlation statistics) are very selective and are missing several variables. Currently, the only significant correlation between Bmax and EEG measures that was found within the PD group and shown in Table 2 is the one between BmaxFC and theta power.
It is important to see correlation statistics between Bmax (both FC and thalamus) and all calculated EEG measures, whether or not significant correlations are found. Of particular interest are the correlations between Bmax (both FC and thalamus) and spectral ratios, which are shown only for the HC but not for the PD group.
Concerning “spectral ratios”, they were defined “as power of alpha+beta bands, divided by power of delta+theta bands”. This definition is very restrictive and may and may lead to false negative conclusions. It is strongly advisable that alternative spectral ratios are calculated and shown. Of particular interest is the alpha / theta ratio, which is commonly used in several EEG spectral power studies. Focusing on alpha and theta without interference by beta and delta may yield more significant associations.
- Claims of possible “causation” based on the current findings are not justified
For example:
Introduction lines 66-68:
“previous studies did not test the specific role of α2 adrenoceptors in changes of EEG and cognitive impairment of PD patients.”
Discussion lines 216-218:
“The inverse correlations indicate that α2 adrenoceptors in part are responsible for the forming of EEG patterns in general, as linked to the slowing of the EEG.”
The current study cannot test a “role” of alpha2 receptors in EEG changes or in cognitive impairment. It just tests for correlations. Causative pathways cannot be reliably inferred from correlation studies. Pharmacological studies with alpha2 agonists and antagonists are better suited to test for possible causative role. The issue of causation vs correlation must be carefully dealt throughout the manuscript.
- Effects of alpha2 agonism & antagonism
Conflicting statements:
Introduction lines 57-58:
“In animal studies, α2 adrenoceptor antagonism raises the EEG spectral ratio (SR) value ... (6)”
Discussion lines 260-261:
“α2 adrenoceptor antagonists lower spectral ratios in rodents (6)”
Please make the appropriate corrections and elaborations on the important matter of alpha2 agonist & antagonist effects on the EEG.
- Significant inverse correlation of BRF with Bmax (in FC and in thalamus) in healthy subjects
A non-significant correlation of the same direction is also noted in the PD group.
This is an interesting finding inadequately discussed. Is it novel ? It is a finding totally different from applying an alpha2 agonist or antagonist to test their effects on EEG. What is known about the correlation between alpha2 receptor density and overall noradrenergic output to cortex and thalamus ?
- Significant positive correlation of age with BmaxFC in healthy subjects
This is an interesting finding (another one in healthy subjects besides the BRF – Bmax correlations). It even made its way to the abstract, but it is not discussed at all. Is this finding novel ? Is there any relevant information available ?
- Additional variable needed: Time since onset of motor symptoms
This is an important variable for PD studies, usually showing stronger, compared to age, correlations to PD parameters. It should be added to Tables 1 and 2 and all relevant correlations should be discussed accordingly.
SPECIFIC POINTS
Abstract
The direction of the correlations must be specified:
Patients’ theta band power was inversely correlated with α2 adrenoceptor density in frontal cortex.
In HC subjects, age and occipital BRF were inversely correlated with α2 adrenoceptor density in frontal cortex, while occipital BRF was inversely correlated with α2 adrenoceptor density in thalamus.
Comments on the Quality of English LanguageMinor editing of English language required
Author Response
Reviewer 3:
GENERAL POINTS
The correlation of noradrenergic system markers with EEG measures in both PD and healthy subjects is interesting and promising. The correlations currently presented are overall marginal and not adequate to strongly favour the presented hypotheses regarding a significant role of the noradrenergic system on EEG spectral characteristics. Certain modifications of the manuscript may help.
While Table 1 suitably contains all variables of interest, Tables 2 and 3 (Correlation statistics) are very selective and are missing several variables. Currently, the only significant correlation between Bmax and EEG measures that was found within the PD group and shown in Table 2 is the one between BmaxFC and theta power.
Author response: We agree that, including more variables in the regression analyses would provide data to fill the tables. We did however not have enough participants to include more variables in the regression analyses. The method for selecting variables for the regression analyses have been provided under method. When using regression analyses, one should not include more than 1 independent variable per 10 observations. We do already violate this, as we have tried to balance providing meaningful information and not including so many variables that the regression would become useless.
Changes to manuscript: Page 6, line 3: “We established the correlation between values and EEG patterns by linear regression with stepwise removal of the least significant explanatory at each step. Explanatories eliminated early were reintroduced to check for confounding. Each analysis began with the inclusion of the variables age, gender, band power of all bands, background rhythm frequency (BRF), and spectral ratios as explanatories.”
It is important to see correlation statistics between Bmax (both FC and thalamus) and all calculated EEG measures, whether or not significant correlations are found. Of particular interest are the correlations between Bmax (both FC and thalamus) and spectral ratios, which are shown only for the HC but not for the PD group.
Author Response: Please see the answer for the comment above. In short, as we should not be including more than one independent variable per 10 observations, our method for choosing the independent variables in the tables are now better described under “Method”, this was not the case in the earlier manuscript, we have now elaborated on this part of the method.
Changes to manuscript: Page 6, line 3: “We established the correlation between values and EEG patterns by linear regression with stepwise removal of the least significant explanatory at each step. Explanatories eliminated early were reintroduced to check for confounding. Each analysis began with the inclusion of the variables age, gender, band power of all bands, background rhythm frequency (BRF), and spectral ratios as explanatories.”
Concerning “spectral ratios”, they were defined “as power of alpha+beta bands, divided by power of delta+theta bands”. This definition is very restrictive and may and may lead to false negative conclusions. It is strongly advisable that alternative spectral ratios are calculated and shown. Of particular interest is the alpha / theta ratio, which is commonly used in several EEG spectral power studies. Focusing on alpha and theta without interference by beta and delta may yield more significant associations.
Author response: We have complied with earlier research in this matter. It may be that focusing on alpha and theta would yield more significant associations. The study is conducted in a manner where we chose the analyses before we began the analytics. We tried to limit the number of analyses due to the limitations put upon us by the limited number of participants. We believed that including more data (all 4 bands) in the ratio would yield a more accurate picture. We can only encourage future research to include more participants and compare different ratios, to find if one is superior.
Claims of possible “causation” based on the current findings are not justified
For example: Introduction lines 66-68:
“previous studies did not test the specific role of α2 adrenoceptors in changes of EEG and cognitive impairment of PD patients.”
Author response: We do not understand this comment. We do not claim any causation in these lines. We merely state that earlier studies have not tested for the role of α2 adrenoceptors in changes of EEG and cognitive impairment of PD patients. This is part of the background for why we want to test the association between α2 adrenoceptor density and changes of EEG.
Changes to manuscript: Under introduction, page 2 line 5 from bottom: “However, previous studies did not test the correlation between α2 adrenoceptor density and changes of EEG and cognitive impairment of PD patients.”
Discussion lines 216-218: “The inverse correlations indicate that α2 adrenoceptors in part are responsible for the forming of EEG patterns in general, as linked to the slowing of the EEG.”
Author response: We do believe that one should be very careful not to mistake correlation with causation, but when EEG patterns are associated with receptor density, we do believe it is very likely that there is a causality between the two, as neuron activity is the source of the measures of EEG, and neuron activity is regulated by neurotransmitters. The causality may not be straightforward and does very likely involve other factors as well. We have however lessened the claim of causation by using “indicate”. We believe this is the right wording in this case.
The current study cannot test a “role” of alpha2 receptors in EEG changes or in cognitive impairment. It just tests for correlations. Causative pathways cannot be reliably inferred from correlation studies. Pharmacological studies with alpha2 agonists and antagonists are better suited to test for possible causative role. The issue of causation vs correlation must be carefully dealt throughout the manuscript.
Author response: We agree that we overstep what we can prove, and that we only investigate correlations. As EEG is a measurement of electrical activity within the brain and this is regulated by neurotransmitters, there is a likely causal relationship between the alpha2 receptors and the EEG measures in the direction alpha2 receptor changes -> EEG changes. But we cannot in any way describe the “role” or how this causality works. We agree that this part of the introduction is misleading.
Changes to manuscript: Page 2, line 6 from the bottom: “However, previous studies did not test the correlation between α2 adrenoceptor density and changes of EEG and cognitive impairment of PD patients.”
Page 9, line 1: “A possible correlation between the power of the theta band and the binding potential of [11C]yohimbine is consistent with the hypothesis that degeneration of LC and deficient NA transmission may contribute to joint alterations of EEG and cognitive impairments in PD.”
Page 8, line 21: “The links may in part may explain how NA denervation of LC may relate to global cognitive performance in patients with PD as shown by means of [11C]MeNER imaging (5). “
Page 7, line 2: “We tested the claim that α2 adrenoceptor density may correlate with EEG patterns.”
Effects of alpha2 agonism & antagonism
Conflicting statements: Introduction lines 57-58: “In animal studies, α2 adrenoceptor antagonism raises the EEG spectral ratio (SR) value ... (6)”
Author response: Thank you, this has been corrected.
Changes to manuscript: Page 8, line 10: “is of interest that the EEG alterations may be direct effects of altered α2 adrenoceptor expression, as found in animals when α2 adrenoceptor antagonists increase spectral ratios in rodents (8).”
Discussion lines 260-261: “α2 adrenoceptor antagonists lower spectral ratios in rodents (6)”
Author response: Thank you, this has been corrected.
Changes to manuscript: Page 8, line 10: “is of interest that the EEG alterations may be direct effects of altered α2 adrenoceptor expression, as found in animals when α2 adrenoceptor antagonists increase spectral ratios in rodents (8).”
Please make the appropriate corrections and elaborations on the important matter of alpha2 agonist & antagonist effects on the EEG.
- Significant inverse correlation of BRF with Bmax (in FC and in thalamus) in healthy subjects. A non-significant correlation of the same direction is also noted in the PD group. This is an interesting finding inadequately discussed. Is it novel? It is a finding totally different from applying an alpha2 agonist or antagonist to test their effects on EEG. What is known about the correlation between alpha2 receptor density and overall noradrenergic output to cortex and thalamus?
Author response: We thank the reviewer for this question. We now cite “Distribution of the Noradrenaline Innervation and Adrenoceptors in the Macaque Monkey Thalamus” by Isabel Pérez-Santos et al.( 2021). In this paper, the authors note that the nuclei with the highest alpha2 receptor densities were found in the lateral dorsal, centromedian, medial and inferior pulvinar and in midline nuclei, suggesting a role for NA in modulating thalamic involvement in consciousness that would support a particular research focus on thalamus, as now cited in the paper.
- Significant positive correlation of age with BmaxFC in healthy subjects. This is an interesting finding (another one in healthy subjects besides the BRF – Bmax correlations). It even made its way to the abstract, but it is not discussed at all. Is this finding novel ? Is there any relevant information available?
Author response: We thank the reviewer for this question. Kuwabara et al. (2012)demonstrated age-associated declines in Bmax in functional striatal subdivisions
where irrespective of methods, KD remained unchanged with age. We now cite this
evidence in the paper.
- Additional variable needed: Time since onset of motor symptoms. This is an important variable for PD studies, usually showing stronger, compared to age, correlations to PD parameters. It should be added to Tables 1 and 2 and all relevant correlations should be discussed accordingly.
Author Response: Thank you, we agree that this would be of great interest in this study. We do however not have this information. This has been added as a limitation under the “Dementia” part of discussion.
Changes to manuscript: Page 9, line 9: “With respect to subgroups of patients, the present study did not permit extended following of patients after imaging after onset of PD. As subgroups are likely to differ in the course of disease, it is important to know the duration of disease when paraclinical data are compared.”
SPECIFIC POINTS
Abstract. The direction of the correlations must be specified: Patients’ theta band power was inversely correlated with α2 adrenoceptor density in frontal cortex. In HC subjects, age and occipital BRF were inversely correlated with α2 adrenoceptor density in frontal cortex, while occipital BRF was inversely correlated with α2 adrenoceptor density in thalamus.
Author response: Thank you for this comment. We have corrected this.
Changes to manuscript: Page 1, line 12: “Patients’ theta band power inversely correlated with α2 adrenoceptor density in frontal cortex. In HC subjects, age correlated with, and occipital background rhythm frequency (BRF) inversely correlated with, α2 adrenoceptor density in frontal cortex, while occipital BRF inversely correlated with α2 adrenoceptor density in thalamus.”
Comments on the Quality of English Language: Minor editing of English language required.
Author response: Thank you for this comment. We have made corrections throughout the manuscript.
Reviewer 4 Report
Comments and Suggestions for Authors
This is an interesting manuscript, however, the low number of patients with PD is a limitation in the Discussion. Also the a lack of cognitive formal evaluation of the PD and HC subjects. In this section, the proposal of longitudinal studies also is not mentioned to validate the findings
In the 2.2. Radiochemistry adds the mechanism of the target of the 11-Cyohimbine to a better understanding to the readers and not only the reference.
Author Response
Reviewer 4:
This is an interesting manuscript, however, the low number of patients with PD is a limitation in the Discussion. Also, the lack of cognitive formal evaluation of the PD and HC subjects. In this section, the proposal of longitudinal studies also is not mentioned to validate the findings. In the 2.2. Radiochemistry, add the mechanism of the target of the 11-Cyohimbine to a better understanding to the readers and not only the reference.
Author response: We have added a limitation part of the discussion. In the conclusion paragraph, we discuss that the current paper may as basis for future studies should be longitudinal and also include a include an exhaustive cognitive testing-battery.
Changes to manuscript: Page 9, line 9: “With respect to subgroups of patients, the present study did not permit extended following of patients after imaging after onset of PD. As subgroups are likely to differ in the course of disease, it is important to know the duration of disease when paraclinical data are compared.
Limitations
The study is limited by the low number of participants that may explain the lack of strong correlation between the value of Bmax measured in thalamus and the power of theta registered in patients. As the design of the study only allowed the search for correlation and not for causation, we believe that future studies should be designed to increase the likelihood of causation and maybe also to more precisely identify subgroups of patients with risk of developing cognitive impairments. The impact of the study is limited also by the complexity of the relations among neurotransmitters because the correlation between EEG and α2 adrenoceptor expression may not be straightforward and may include multiple mechanisms and neurotransmitters that we cannot in any way rule out in the present study.”
Round 2
Reviewer 3 Report
Comments and Suggestions for Authors
Comments have been addressed adequately and the manuscript is improved.
I have 2 methodological comments on the authors' response:
Concerning the limited number of EEG metrics listed in Tables 2 and 3
Authors’ response is unclear. It is still unclear why, for example, Table 3 – BmaxFC contains the beta power model, but not the delta power model, whereas Table 3 – BmaxThalamus contains the delta power model, but not the beta power model. It does not appear, for example, that the BmaxFC beta power result was selected by a stepwise process, since it seems to be virtually totally unrelated to BmaxFC (Coef=0.031, P-value=0.902). In my opinion, Tables 2 and 3 would be more helpful to readers if each one of the 4 data columns contained all EEG metrics listed in Table 1 and tested in statistical models.
Concerning the alpha-beta and theta-delta spectral power pooling
There is no rule obliging researchers to “comply” by pooling alpha-beta and theta-delta spectra. Pooling is often used as a screening step, followed by more specific metrics that look promising. Surely “more comparisons” are more likely to yield statistical significance, but truly strong associations can be obtained, and should be sought for, even with small numbers of participants.
Author Response
Author: We thank you for your time and comments, we found the first round of comments to improve the quality of paper greatly. We are grateful that you continue to help us improve this paper.
- Concerning the limited number of EEG metrics listed in Tables 2 and 3. The authors’ response is unclear. It is still unclear why, for example, Table 3 – BmaxFC contains the beta power model, but not the delta power model, whereas Table 3 – BmaxThalamus contains the delta power model, but not the beta power model. It does not appear, for example, that the BmaxFC beta power result was selected by a stepwise process, since it seems to be virtually totally unrelated to BmaxFC (Coef=0.031, P-value=0.902). In my opinion, Tables 2 and 3 would be more helpful to readers if each one of the 4 data columns contained all EEG metrics listed in Table 1 and tested in statistical models.
Aurthor response: Thank you for this comment. We agree that the statistics should be as stringent as possible and that this should be reflected in the tables. In our study we are however limited by the low number of participants. This has led us to have to choose between consistency in method or consistency in presentation. We chose to be stringent in method. Tables 2 and 3 are the results of regression analyses. Each analysis began with the inclusion of the variables age, gender, band power of all bands, background rhythm frequency (BRF), and spectral ratios as explanatories. This means that they were done in the same manner, and we ended with the number of variants we believe the analyses could contain and still be reliable (because the number of variants that a regression analysis can contain depends on the value of N). As described under method, we used linear regression with stepwise removal of the least significant explanatory at each step and explanatories eliminated early were reintroduced to check for confounding. We believe this is the statistical correct way. This also means, that variables not included in the models are not significant (And would not show significance in later steps of the regression analyses, as this was also checked for by the reintroducing of the variable). All variables were chosen by this method regardless of how it appears to this reviewer. We edited the method sections to further explain this point.
Changes to manuscript: Page 5, line 5 under ”Method”: “variables of age, gender, band power of all bands, background rhythm frequency (BRF), and spectral ratios as explanatories. We removed variables due to the limited number of participants in our study, but did not remove variables having significant impact.”
- Concerning the alpha-beta and theta-delta spectral power pooling: There is no rule obliging researchers to “comply” by pooling alpha-beta and theta-delta spectra. Pooling is often used as a screening step, followed by more specific metrics that look promising. Surely “more comparisons” are more likely to yield statistical significance, but truly strong associations can be obtained, and should be sought for, even with small numbers of participants.
Author response: We agree that the literature is not completely stringent in this matter, and our last response may have been poorly worded. This is a case in which we compare the risk of committing a type 2 error with the risk of committing a type 1 error. We do not believe we have the number of participants to defend this many comparisons. We believe that our result can be used in meta-analyses to find the “truly strong associations”. We too believe that there is an association and hope to inspire others to conduct this meta-analysis.